# Volatile organic compound patterns predict fungal trophic mode and lifestyle

Yuan Guo[1], Werner Jud[1], Fabian Weikl[2], Andrea Ghirardo [1], Robert R. Junker[3,4], Andrea Polle [5,6], J. Philipp Benz[7], Karin Pritsch[2], Jörg-Peter Schnitzler [1] & Maaria Rosenkranz [1✉]

Fungi produce a wide variety of volatile organic compounds (VOCs), which play central roles in the initiation and regulation of fungal interactions. Here we introduce a global overview of fungal VOC patterns and chemical diversity across phylogenetic clades and trophic modes. The analysis is based on measurements of comprehensive VOC profiles of forty-three fungal species. Our data show that the VOC patterns can describe the phyla and the trophic mode of fungi. We show different levels of phenotypic integration (PI) for different chemical classes of VOCs within distinct functional guilds. Further computational analyses reveal that distinct VOC patterns can predict trophic modes, (non)symbiotic lifestyle, substrate-use and host-type of fungi. Thus, depending on trophic mode, either individual VOCs or more complex VOC patterns (i.e., chemical communication displays) may be ecologically important. Present results stress the ecological importance of VOCs and serve as prerequisite for more comprehensive VOCs-involving ecological studies.

[1] Research Unit Environmental Simulation, Institute of Biochemical Plant Pathology, Helmholtz Zentrum München, Neuherberg, Germany. [2] Institute of Biochemical Plant Pathology, Helmholtz Zentrum München, Neuherberg, Germany. [3] Evolutionary Ecology of Plants, Department of Biology, Philipps University of Marburg, Marburg, Germany. [4] Department of Biosciences, University of Salzburg, Salzburg, Austria. [5] Forest Botany and Tree Physiology, University of Göttingen, Göttingen, Germany. [6] Beijing Advanced Innovation Center for Tree Breeding by Molecular Design, College of Biological Sciences and Technology, Beijing Forestry University, Beijing, People's Republic of China. [7] Holzforschung München, TUM School of Life Sciences Weihenstephan, Technical University of Munich, Freising, Germany. ✉email: maaria.rosenkranz@helmholtz-muenchen.de

Fungi are key components in various ecosystems[1–3]. They have evolved diverse relationships with plants and other organisms and can be grouped into functional guilds such as mycorrhiza[4,5], pathogens[6,7], mycoparasites[8], or saprotrophs[9–11]. These fungal functional guilds have been characterized by various traits encompassing genetic, enzymatic, morphological, and physiological metrics[12]. For example, free-living filamentous fungi are more likely to possess traits related to decomposition. In contrast, such traits might lack in symbiotic fungi, such as mycorrhizae, that generally obtain their carbon from the host[13]. Though trait-based approaches have provided diverse perspectives to describe the multifold ecological functions of fungi, these approaches are far from covering all fungal functions in ecosystems[12,14]. It is, e.g., difficult to define the transition of individual species among different guilds and lifestyles. Many fungal species are opportunistic and change their lifestyle depending on environmental conditions[12,13]. Fungal taxa are, moreover, not functionally equivalent in their contributions to different traits[13]. The development of new function-related fungal metrics can improve our understanding of fungal functions in nature, and in combination with novel approaches, they may help to define different functional guilds.

Fungi and other micro-organisms emit a wealth of highly diverse volatile organic compounds (microbial VOCs; mVOCs)[15–19]. Compared to the more common morphological, physiological, and biochemical traits, understanding the causes why and when fungi release specific VOCs is in its infancy. The limited knowledge is, on the one hand, due to restrictions in the methods enabling the detection of only parts of the fungal VOC (fVOC) spectra[18], but also due to the limited knowledge about the biological and ecological functions of individual VOCs and their mixtures[19,20]. An increasing number of studies suggest crucial direct and indirect functions for fVOCs in fungal interactions[20–23]. Direct antimicrobial properties were revealed for VOC patterns of several fungi, such as myco-parasitic *Trichoderma* spp.[24], endophytic *Muscodor albus*[25], or non-pathogenic *Fusarium oxysporum*[26]. Also, plant performance can directly be impaired as shown for fVOC blends emitted by the phytopathogens *Cochliobolus sativus* and *Fusarium culmorum*[27]. Most research on individual fVOCs has so far focused on small molecules such as short-chain alcohols, ketones and aldehydes[28–30]. The small fVOC, 1-decene, e.g., can alone induce the growth of *Arabidopsis* plants and alter the expression of several defense and stress-related genes[28]. In contrast, other small VOCs such as 2-methylpropanol and 3-methyl-butanol released by *Phoma* spp., enhance the performance of tobacco plants only in mixtures but not individually[31]. Though such small compounds may be by-products of the primary fungal metabolism[32,33] and do not exhibit high species or fungal guild specificity, they may still—alone or in mixtures—have important functions in fungal interactions[34]. Considering larger, semi-volatile compounds, the sesquiterpene, (-)-thujopsene released by the mycorrhizal fungus *Laccaria bicolor* induced lateral root growth of the host plant in the pre-colonization phase, thus facilitating the formation of symbiosis[35]. Also, other individual fVOCs, such as 6-pentyl-α-pyrone from *Trichoderma* spp. were shown to alter plant performance[36].

Although some knowledge exists about the potential functions of individual compounds or compound mixtures[37], no attempt has been made to link comprehensive VOC profiles—the volatilomes—to the ecological functions and lifestyles of individual fungal species. Previously, we demonstrated that trophic modes of forest fungi—pathogens and saprotrophs and ectomycorrhizal fungi—can be distinguished by their volatile patterns[38]. However, due to the limited number of species (eight) it remained at that time unrevealed whether and to what extent the chemical diversity of fVOCs can be associated with different ecological functions on a broader scale.

The use of fVOCs in chemotaxonomy has been widespread[39–41]. Traditionally, mushroom fruitbodies' typical flavors are used in species identification during mushroom picking[42]. Species-specific volatile biomarkers have also been used as indicators of harmful fungi in industry or different indoor environments[39,43,44]. Moreover, based on the fVOC patterns, fungi can be classified taxonomically at the level of genera[39,40,45,46] and families[47]. VOC-based chemotaxonomy can potentially complement traditional fungal identification schemes[48]. However, before broader, fVOC-based taxonomic or functional guild-based characterizations are possible, more comprehensive and systematic VOC measurements across several phyla and fungal guilds are required.

In this study, we characterized the volatilomes from mycelia of 43 fungal species cultivated under standardized conditions. To be able to make broad and valid statements, we selected representative species from the three phyla Ascomycota, Basidiomycota, and Zygomycota (sensu Spatafora et al.)[49] and included different ecological guilds (such as trophic mode, lifestyle, substrate-use, and host species). The fungal volatilomes were determined employing a novel platform[18] that combines an automated cuvette system and two mass spectrometric methods, the proton transfer reaction mass spectrometry (PTR–MS) and gas chromatography–mass spectrometry (GC–MS). This approach enabled the characterization of species-specific fungal volatilomes, and their classification in phylogenetic and functional relationships. Following Junker et al.[50], we analyzed the phenotypic integration (PI) of fVOCs in the different fungal guilds and chemical structural classes. PI was previously used to reveal the integration level within chemical communication displays (CCDs), i.e., complex VOC patterns used for intra- and inter-specific communications[50]. Covariation between quantitative traits, which are fVOCs in this study, leads to high PI and indicates that fixed proportional compositions of VOC bouquets may be required to achieve optimal interactions. Finally, using a custom-built machine learning strategy we uncovered the volatile biomarkers underlying functional groups and thus provide essential fundamentals for further ecological validation. Altogether, our analyses revealed fundamental links between volatile profiles and the ecological functions of fungi.

## Results

**Fungal VOC profiles—species-specific fingerprints.** We detected a total of 256 volatile compounds from 43 fungal species (Fig. 1, for the fungal species please refer to Supplementary Table S1[51], for the individual VOCs please see Supplementary Tables S2 and S3[51]). In hierarchical dendrograms, clusters comprising VOCs emitted by several species (e.g., cluster A, D, F, H, J, P) but also species-specific fingerprints (e.g., cluster B, C, E, G, I, K, L, M, N, O, Q) were identified. Cluster H comprises the largest group of compounds consisting of rather common VOCs, such as various alcohols, emitted by most fungal species. In addition to this more common volatiles, we found characteristic clusters for individual species. The largest species-specific clusters were cluster G, specific for the mycoparasitic *Trichoderma reesei*, cluster Q specific for the phytopathogen *Bipolaris sorokiniana* and cluster M specific for the phytopathogen, *Heterobasidion annosum*. Other smaller species-specific clusters were found: I for *Amanita porphyria*, K for *Alternaria brassicicola*, and E for *Trametes hirsuta* (Fig. 1).

In general, a higher number of compounds were detected by PTR–MS than by GC–MS from individual species (Fig. 2), but the fVOC patterns detected by GC–MS showed higher variability than the patterns detected by PTR–MS. With PTR–MS especially short-chained alkenes and other carbonyl compounds, that were

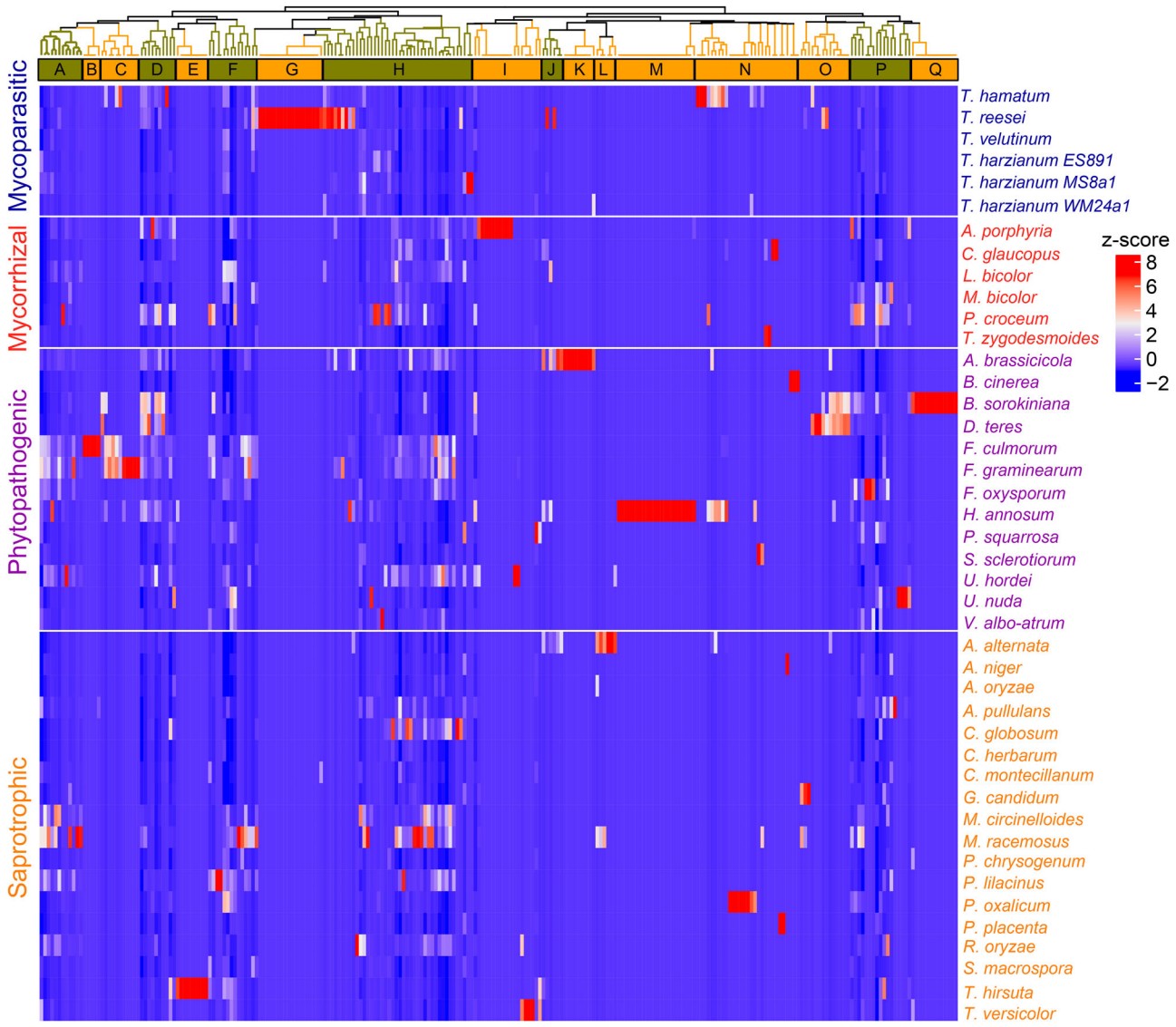

**Fig. 1 Heatmap of fungal volatile organic compounds (fVOC) emission profiles from the examined 43 fungal species.** The dendrogram shows the hierarchical clustering based on Spearman correlations ($\rho < 0.05$). Compounds are grouped to 17 clusters (letters A-to-Q). The letters refer to the groupings in the Supplementary Tables S2 and S3[51]. The emission rates (ncps cm$^{-2}$ s$^{-1}$ and pmol cm$^{-2}$ h$^{-1}$) based on PTR–MS and GC–MS data, respectively, are color-coded: red indicates high and blue indicates low emission.

not possible to measure by the GC–MS set-up, were detected. In contrast, with the Twister/GC–MS-combination mainly sesquiterpenes and other structurally more complex volatile compounds were measured (Fig. 2c and Supplementary Tables S2, S2[51]).

To evaluate the chemical diversity of compounds, we applied Pilou evenness ($J$) to measure the distribution of compounds in individual species ($J$ is constrained between 0 and 1.0, where $J = 1$ means all the compounds were emitted in equal abundance). In general, the GC–MS detected compounds show higher $J$ compared to compounds detected by PTR–MS in each species (Fig. 2a, e). Some species have relatively low $J$ such as *L. bicolor*, *Cladosporium herbarum*, *Postia placenta* and *Penicillium oxalicum*, suggesting their emission profiles are dominated by individual or few compounds (Fig. 2a, e). The emission intensities showed high variation across all the species (Fig. 2b, d and Supplementary Fig. S1). Particularly high intensities were detected by PTR–MS from the mycorrhizal fungi *A. porphyria* and *L. bicolor* as well as from the phytopathogen *Ustilago hordei*. Emissions from these species were characterized by high levels of low molecular weight VOCs including some carbonyl compounds

(e.g., *m/z* 87.081, pentanal or pentenol) (Fig. 2c and Supplementary Fig. S1, Supplementary Table S4[51]). GC–MS revealed high emission intensity, especially from the two mycorrhizal fungi *Piloderma croceum* and *A. porphyria*. The high emission intensities were not especially associated to low or high evenness (Fig. 2a, b, d, e).

**VOC profiles allow characterization of fungal guilds in the level of phyla.** Discriminant analysis of the main components (DAPC) of the complete emission profiles showed a clear separation between the different phyla, i.e., Ascomycota, Basidiomycota, and Zygomycota (Fig. 3a). In respect to the taxonomic class level, the volatile emissions of Ustilaginomycetes and Zygomycetes strongly differed from the other classes (Fig. 3b). On the level of orders, Atheliales, Ustilaginales and Pleosporales had the most distinct VOC profiles (Fig. 3c). When the fungi were grouped by the taxonomic level family, the volatiles emitted by Amanitaceae and Atheliaceae showed a pronounced diversity compared to those from others (Fig. 3d).

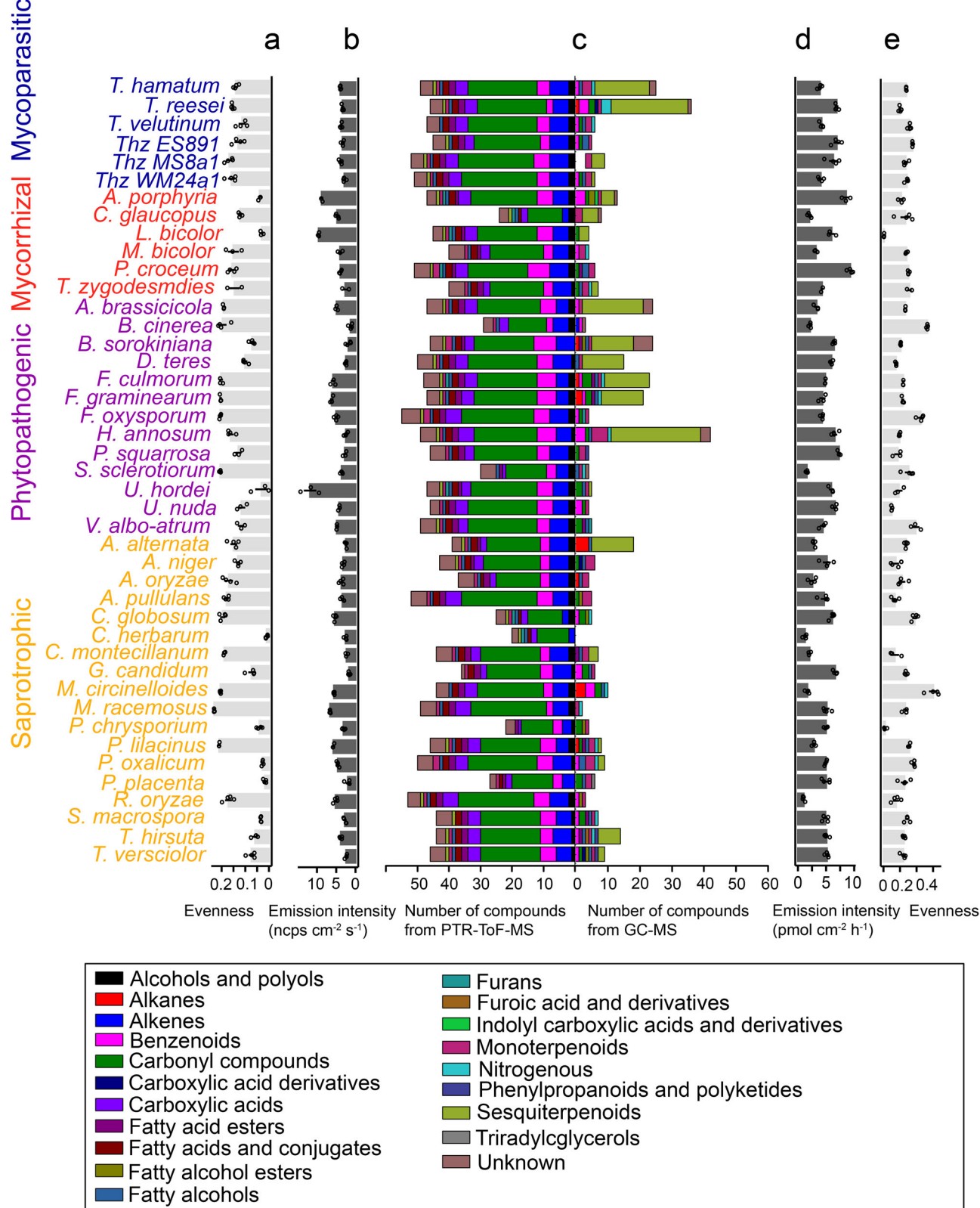

**Fig. 2 Chemical diversity of fungal volatile organic compounds (fVOCs) detected from the examined 43 fungal species. a, e** The Pilou evenness of species-specific fVOC profiles based on PTR-ToF-MS and GC–MS data, respectively. **b, d** The total emission rates (ncps cm$^{-2}$ s$^{-1}$ and pmol cm$^{-2}$ h$^{-1}$) based on PTR-ToF-MS and GC–MS data, respectively and **c** show the number of compounds detected by PTR-ToF-MS and GC-MS. Colors indicate the grouping of fVOCs to structural classes. Thz *T. harzianum*. Data are shown as means ± SE (*n* = 3 biologically independent samples).

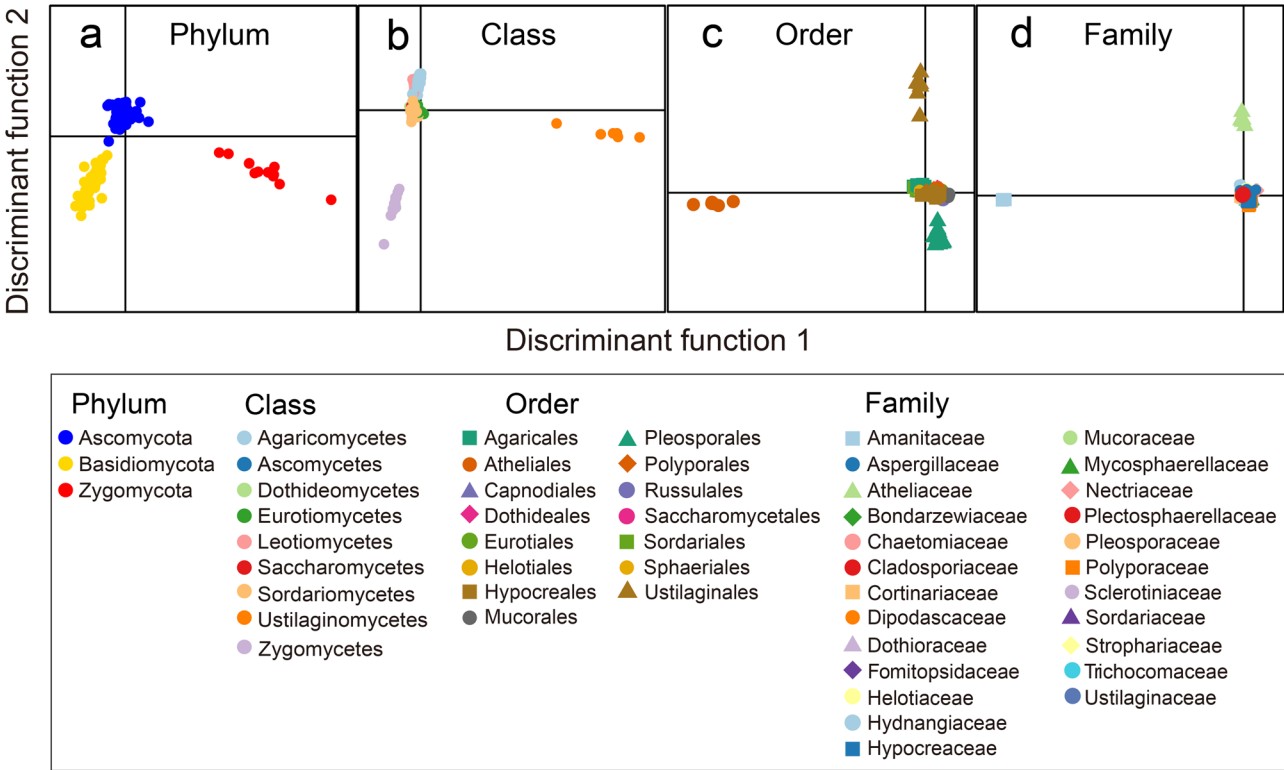

**Fig. 3 Chemotaxonomic characterization of different fungal clades.** The chemotaxonomic characterization comprises phylum (**a**), class (**b**), order (**c**), and family (**d**) by discriminant analysis of principal components (DAPC) using combined PTR-ToF-MS and GC–MS data. Detailed parametric configurations of DAPC models are provided in the "Methods" section.

**Trophic modes, substrate-use, and fungal lifestyle are associated with certain scents.** We investigated whether the trophic mode, substrate-use and symbiotic or non-symbiotic lifestyle of the different fungi can be related to the emission patterns and scents. When the species were grouped in most common guilds based on literature (Supplementary Table S1[51]), phytopathogens and mycorrhizal fungi showed distinct VOC profiles (for all emission intensities in ncps and in pmol (cm$^{-2}$ h $^{-1}$), see Supplementary Table S4[51]) compared to saprotrophic and mycoparasitic fungi and were therefore clearly separated in the DAPC analysis (Fig. 4).

Saprotrophic and mycoparasitic species exhibited largely overlapping chemical profiles (Fig. 4a). Similar VOC profiles might reflect the ability of some of the species to easily move among trophic modes. It might be that the fungi that easily changes its trophical mode, might show a flexible, environmentally adjustable VOC profile that fits to more than one guild. This is supported by the second DAPC analysis (Supplementary Fig. S2) in which we used an alternative grouping of the fungi that possess an obvious alternative trophical mode (i.e., *Trichoderma* spp. was moved from mycoparasites to saprophytes; *Alternaria alternata* and *Aspergillus niger* from saprophytes to phytopathogens and *Fusarium oxysporum* from phytopathogens to saprophytes; for details of the alternative guild and related references please see Supplementary Table S1[51]). The results on this alternative grouping still reveal a clear separation of phytopathogens, saprotrophes, and mycorrhizal fungi by their volatile profiles (Supplementary Fig. S2).

Though ecological functions explained only part of the volatile profiles, the fungal lifestyles could generally be described by different chemotypes: Non-symbiotic fungi showed a significantly different volatile chemotype under the chosen growth conditions compared to fungi with symbiotic lifestyle (Fig. 4b). Root-associated fungi (in this study mycorrhizal and root pathogenic fungi) could also be clearly separated from fungi that live on litter or from fungi associated with shoots (Fig. 4c). An evident separation was also found between the VOC chemotypes of herb- and tree-associated fungi (Fig. 4d).

The calculation of PI revealed high values within specific structural classes of compounds (Fig. 5). The highest PI-values were revealed for alkanes released by phytopathogens and saprotrophs, carboxylic acids released by saprotrophs and mycorrhizal fungi and fatty alcohols released by mycorrhizal fungi. Other significant PI-values were found for monoterpenoids released by saprotrophs (Fig. 5 and Supplementary Table S5[51]). Across all detected compounds, the mycoparasites showed higher PI than the other studied trophic modes, however, the integration within the CCDs was not significant. Especially low PIs were revealed for different esters and other carbonyl compounds (i.e., aldehydes and ketones) within all the eco-functional groups.

**Volatile biomarkers of phylogenetic groups and functional guilds.** We identified discriminatory VOC biomarkers and chemical profiles that are characteristic for individual taxonomic guilds, trophic modes, and lifestyles of fungi. To do that we have applied a strict, unbiased machine learning strategy. It aimed at identifying a minimal set of VOCs which can be used to describe taxonomic assignments and the classification into trophic modes and life forms, without a pronounced loss of predictive accuracy. At the end, we were able to use the 15 best predictors to describe each fungal group with an accuracy of more than 80%. The detailed model of each class is presented in Table 1. Among the three phyla present in our study, Zygomycota were described by strong emission of multicomponent volatile profiles (comprising H30 (*m/z* 73.065), H27 (*m/z* 71.049), H12 (*m/z* 81.034),

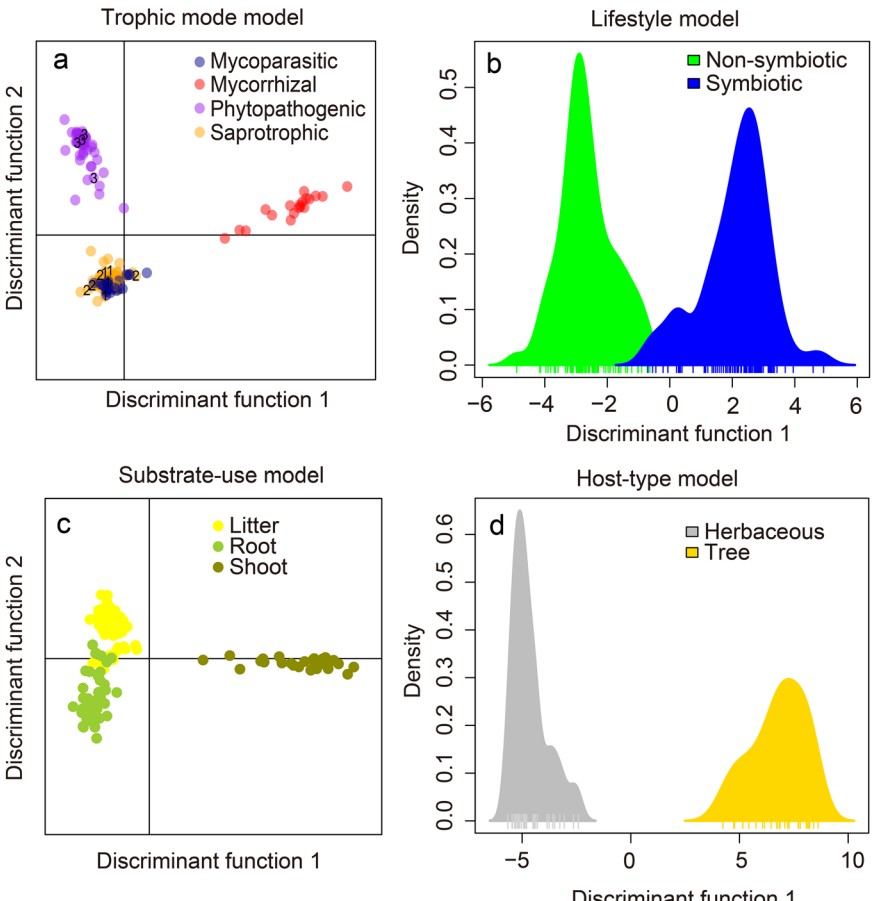

**Fig. 4 Functional characterization of fungi using the complete fungal volatile organic compound (fVOC) profiles.** The discriminant analysis of principal components (DAPC) models comprise trophic mode (**a**), (non)symbiotic lifestyle (**b**), substrate-use (**c**), and host-type (**d**) models. **a**, **c** Scatter plots with dots denoting individual fungus; **b**, **d** Density plots, with ticks denoting individual fungus. The numbers in **a** indicate the fungal species easily moving among trophic modes (*Alternaria alternata* (#1) and *Aspergillus niger* (#2)) were grouped to saprotrophs, *Fusarium oxysporum* (#3) to phytopathogens (for details refer to Supplementary Table S1[51]). Detailed parametric configurations of DAPC models are provided in the "Methods" section.

A5 (*m/z* 105.092), H34 (*m/z* 63.045), A1 (*m/z* 43.055), H40 (*m/z* 53.039), A6 (*m/z* 91.076), A4 (*m/z* 41.038), F15 (γ-collidine), H10 (dodecyl acrylate), F14 (4-ethylresorcinol), F10 (phenol), I16 (putative diethyl phthalate), F11 (*cis*-hexahydrophthalide) and L3 (hexadecane)). Basidiomycota were described by ten different compounds (F9 (*m/z* 85.065), F8 (*m/z* 85.102), P9 (*m/z* 67.055), P8 (oxime-methoxy-phenyl-), H16 (1-octen-3-ol), H38 (3-heptanone,6-methyl-), E1 (ylangene), I20 ((1R)-(+)-trans-iso-limonene), H19 (3-octanone) and I21 (methyl furoate)), and Ascomycota by a relatively strong emission of only two compounds (A10, *m/z* 93.092 and F14, 4-ethylresorcinol). Interestingly, biomarkers characteristic for symbiotic fungi were emitted in high amount (D1, *m/z* 123.117; H39, *m/z* 57.070; D6, *m/z* 109.101; D5, *m/z* 135.116; H32, *m/z* 47.049; A10, *m/z* 93.092; H33, *m/z* 45.034; D10, *m/z* 33.034; H18, *m/z* 129.128; P8, oxime-methoxy-phenyl-; H16, 1-octen-3-ol; C3, dihydrocurcumene; H38, 3-heptanone,6-methyl-; H4, β-bisabolene; F14, 4-ethylre-sorcinol; H1, acrodinene; P7, Furan,2-pentyl-; J2, α-bergamotene; N18, unknown #1), while non-symbiotic fungi could be described by a low emission of these compounds (Fig. 6). Only four oxygenated compounds, i.e., 4-ethylresorcinol, 3-heptanon,6-methyl-, oxime-methoxy-phenyl- and 1-octen-3-ol, were present uniformly in all groups (Fig. 6 and Supplementary Fig. S2).

We were able to identify 13 unique biomarkers (A6, H12, P9, L3, I21, H27, F10, A1, I16, H10, F11, F15, E1), which allow the phylogenetic differentiation of the fungi, at least at the phylum level. Six VOCs allowed us to assign the fungi to their trophic

modes (N16, H3, P1, A2, J3, and H29). Five compounds (D1, D6, D9, N18, and H4) could be associated with lifestyle. A further four compounds (J1, H22, P12, and M22) and finally two compounds (H34 and H21) allowed the identification of host type and substrate-use, respectively (Supplementary Fig. S3 and Supplementary Table S6[51]).

## Discussion

Volatile organic compounds are increasingly recognized as biologically active molecules with a wide range of ecological functions[17,21,23]. Previous literature indicates that fVOC emissions may be species- or genus-specific[38,39,43,46,52]. However, the lack of comprehensive studies that present the full emission spectra and the large structural chemical diversity of these compounds have hampered more complex statistical analyses and phylogenetic comparisons. Difficulties arise from restrictions in analytical techniques, a limited number of studied species, and varying experimental conditions such as physiological stage of the organisms, sampling period, nutrient availability, and other environmental factors such as pH or temperature, that all can influence the fungal performance and the fVOC patterns[53–56]. These constraints make comparisons between different publications difficult and, moreover, it is rather impossible to draw an integral picture of the chemical diversity based on the existing fVOC data. Here we aim to systematically close this gap and establish a link between chemotypic variability of fungal volatiles,

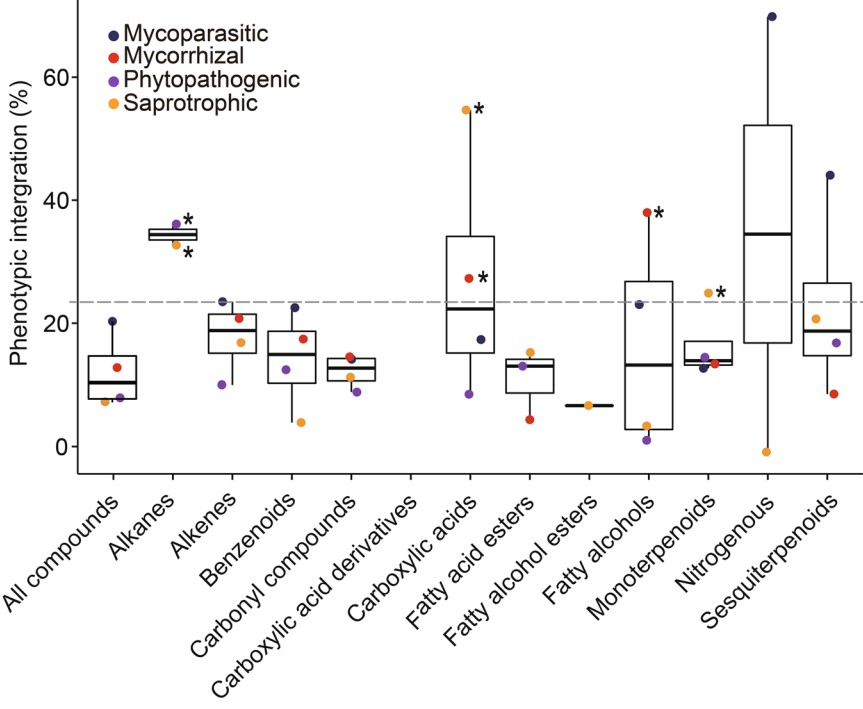

**Fig. 5 Phenotypic integration (PI) of the chemical communication displays (CCDs) of fungi.** The PIs are shown for different structural compound classes in different fungal functional guilds. High integration values indicate a strong covariation between compounds within the structural classes. The horizontal gray dashed line indicates the NULL-model expectation (two-sided null-hypothesis). Asterisks indicate a significant difference ($p < 0.05$) compared to the NULL-hypothesis. The varying sample sizes (number of fungal species within trophic modes) were corrected as shown in the "Methods" section.

**Table 1 Model performance of fungal volatile organic compounds (fVOCs) emission-based models predicting fungal phyla, trophic mode, lifestyle, substrate-use, or host type.**

| | Accuracy (%) | Sensitivity (%) | Accuracy *p*-value of model |
|---|---|---|---|
| Phylum | | | |
| Ascomycota | 94 | 95 | 2.57E−03 |
| Basidiomycota | 93 | 90 | |
| Zygomycota | 100 | 100 | |
| Trophic mode | | | |
| Mycoparasitic | 95 | 92 | 2.19E−08 |
| Mycorrhizal | 94 | 88 | |
| Phytopathogenic | 92 | 84 | |
| Saprotrophic | 94 | 99 | |
| Lifestyle | | | |
| Symbiotic | 91 | 95 | 1.21E−03 |
| Non-symbiotic | | | |
| Substrate-use | | | |
| Litter | 92 | 96 | 7.07E−07 |
| Root | 93 | 86 | |
| Shoot | 93 | 90 | |
| Host-type | | | |
| Tree | 96 | 92 | 8.307E−03 |
| Herbaceous | | | |

Accuracy, sensitivity and model *p*-value are given as the mean of the three predictive models on PTR-ToF-MS and GC–MS data.

large-scale fungal taxonomy, and their trophic modes. As a basis for that served the present, comprehensive volatilome analyses from 43 fungal species cultivated under identical, controlled conditions. The statistical analyses revealed a clear link between complex volatile patterns and specific trophic modes and lifestyles. This was independent of the taxonomic classification of species in the phylogenetic fungal tree of life, and thus in line with the fact that trophic modes of fungi are not monophyletic[13].

Central outcome of our study is the connection between fungal emission patterns and trophic modes. The results suggest that fungal guilds and lifestyles can be grouped according to the fVOC profiles. Knowing that some fungi are able to change their trophic modes, care should, however, be taken when developing, for example, VOC-based non-invasive identification means for different guilds. Some fungi, such as *A. alternata* and *Trichoderma* spp. are known for their multiple trophic modes (please see Supplementary Table S1[51] for details and references). When these species were grouped in their alternative mode, the result from DAPC analysis did not drastically change. This reflects the fact that the secondary metabolism of the species especially known for their multiple trophic modes is not fixed to any specific mode. It is possible that fVOC profiles can change at different environmental conditions[18,41,56]. Thus, it is tempting to suggest that a fungus might also adjust its VOC emission pattern accordingly when adjusting its trophic mode. To better understand the robustness of lifestyle- and trophic mode-related fVOC patterns shown here, we encourage analyzing fVOC profiles of the same species in different environments in the future. The patterns described here can serve as a basis for such further analyses.

At the species level, our results show a high compositional and quantitative variation in fVOCs, which supports the results of the

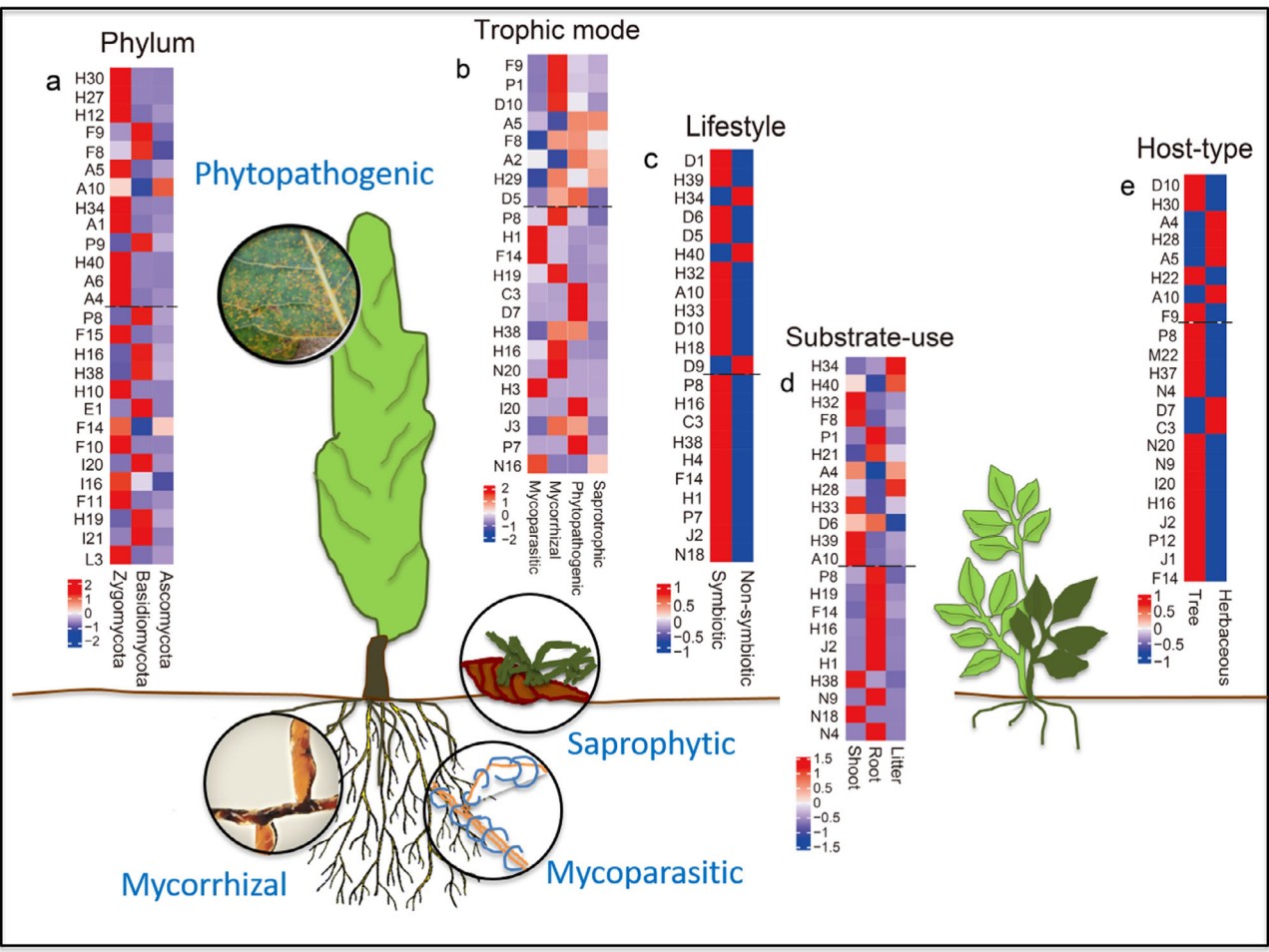

**Fig. 6 Volatile biomarkers identified for phylum, trophic mode, (non)symbiotic lifestyle, substrate-use, and host type.** The dashed line dividing the heatmap separates compounds detected by PTR-ToF-MS (upper part) and GC–MS (lower part) (**a–e**). The compounds are shown in descending order according to its importance from top for PTR-ToF-MS (upper part) and GC–MS (lower part), respectively. The color code refers to the emission intensity (data show z-scores). The letters refer to the compounds listed in Supplementary Tables S2 and S3[51]. The schematic tree and soil scenario illustrate the natural plant–fungi interaction contexts; photo: J-P Schnitzler.

previous studies[43,57,58]. The identified species-specific volatile fingerprints, such as cluster "E" for *T. hirsuta*, cluster "G" for *T. reesei,* or cluster "M" for *H. annosum,* etc., indicate that fVOCs may have potential as non-invasive identification markers. With cluster analyses, we aimed to further disentangle the chemical structure and diversity of fVOCs at different taxonomic levels. We observed characteristic VOC patterns at the phylum level, while at lower taxonomic levels, class, order, and family, only a few groups allowed an assignment via distinct volatile chemotypes. The data thus suggest that the biochemical diversity of synthesized and emitted fVOCs may not explicitly reflect phylogenetic relationships of fungi. Nevertheless, the data reliably allowed to mathematically uncover general taxonomic characteristics in the fungal volatile profiles and are a sound basis for further hypotheses on the chemical taxonomy of fungi.

Regarding potential biological functions of guild- and taxa-related fVOCs, sesquiterpenes are a prominent group of molecules with very high structural diversity[59]. We found 68 and 41 sesquiterpenes emitted by Ascomycota and Basidiomycota species, respectively. In our study, we additionally report a low sesquiterpene emission for *Rhizopus oryzae*, belonging to Zygomycota. Sesquiterpenes may have several functions in fungal interactions

including, e.g., the attraction of pollinators[60], modulation of fungal development[61], mediation of fungal–insect–plant interactions[62,63], and regulation of plant and microbial growth[35,64]. Assigning biological functions to individual compounds in nature is more complicated, as many of the sesquiterpenes released by fungi can also be synthesized by plants or bacteria[65–67]. Many of the other individual VOCs studied to date have, moreover, been shown to exhibit multiple functions[17,23]. For example, 1-octen-3-ol, activates plant defense genes[30] (at low concentrations, while at high concentrations plants get damaged[68]), attracts insects[69], and can inhibit fungal development[70]. 1-Octen-3-ol is a typical fungal VOC emitted by many species from different functional guilds[17]. In the present study, 1-octen-3-ol was detected from nineteen of the studied species.

PI is often used as a tool to infer functional adaptations and physiological limitations from covariation patterns between traits in complex phenotypes[71]. In the present work, distinct PIs were found for different structural classes of compounds within different fungal guilds. For example, saprophytes were characterized by relatively high PI within carboxylic acids and monoterpenoids, whereas mycorrhizal fungi showed high PI within carboxylic acids and fatty alcohols. The VOCs from these specific structural

classes might simply be linked to the substrate availability for the specific biosynthetic routes. Alternatively, these results may indicate that within these groups defined compound classes (chemical communication displays; CCDs), rather than individual VOCs are biologically active. Such CCDs were revealed for example by Naznin and colleagues[31] for small fVOCs that were biologically active only in specific mixtures. The relatively high PI value detected for mycoparasites across all compounds and within specific structural classes might, however, also be partly caused by the fact that this group included only *Trichoderma* spp. In contrast to mycoparasites, the saprotrophic and phytopathogenic fungi had especially low PI. Low PI may indicate that VOCs or/and CCDs do not play important ecological roles for these groups, or individual VOCs are sufficient signaling cues.

In general, whether it is individual compounds, complex VOC patterns, or even silence, fungal VOCs may contribute considerably to the survival of fungi in different ecosystems, and to their adaptation to different environmental conditions[17,72,73]. The adaptability of CCDs,—the universal language of volatile communication—is a prerequisite for the function of VOCs as infochemicals in the interaction between various organisms. This should, however, be also considered when drawing more general pictures from fVOC profiles. Certain genes that regulate and control the biosynthesis of VOCs may be switched off in monoculture, while the same fungus in mixed culture[74] or under natural conditions (e.g., in soil) in interaction with other organisms may have completely different emission patterns. Such interactions include also fungal endobacteria, which are ubiquitous[75] and have been shown to affect the detected fVOC patterns[76]. An additional layer of complexity is set by the dose-dependency of effective VOC signaling cues[77]. Similar to plants[78–80], there is growing evidence that CCDs beside individual compounds may play important roles in the formation and regulation of symbiotic associations and the distribution of saprophytic, mycorrhizal, and phytopathogenic fungi[17,68,81]. Though a lot is still to be done, our analysis represents an important step in fungal ecology. By presenting the distinct, fungal functional guild-related VOC patterns, the present study provides a solid basis for the future use of more natural and realistic VOC bouquets in various ecological studies, be it in the laboratory or in more natural environments.

Using a ML approach, we were able to identify group-specific patterns as well as function- and taxa-related fingerprints. A small number of VOCs of different biochemical origin were sufficient to predict each category. For example, a total of 22 compounds were needed to predict a phylum. In addition, we took advantage of ML to identify the most important functional volatile fingerprints of each functional guild and for the fungal lifestyle. Although there is still a long way to go from the findings of our laboratory study with pure cultures to causal relationships in nature, the characteristic fingerprints described herein provide the basis to form novel hypotheses for future field studies. The current analyses make it very clear that complex volatile patterns of substances from different metabolic pathways can be assigned to distinct trophic modes and lifestyles. The existence of innate VOC patterns or CCDs that characterize ecological functions across taxonomic boundaries is supported by the observation that symbiotic and non-symbiotic fungi can also be categorized by different volatile profiles. Thus, even across different phyla, fungi that share a similar ecological function show intrinsic similarities in their VOC patterns. Different VOC chemotypes can, moreover, be found within the group of symbiotic fungi living on different plant organs. These data underline the need to further investigate the ecological functions of complex, more natural VOC patterns rather than those of single compounds. The individual compounds may have different functions on their own, but these

could be dramatically adjusted in a correct VOC bouquet. In future synthetic CCDs could be used to test whether, for example, plants can distinguish between the scent of a phytopathogenic, saprophytic, or mycorrhizal fungus. The focus of future investigations should lie on the analysis of the robustness of the emission patterns described here, especially for those fungal species that potentially move between different trophic modes. Fungi are highly adaptable and even if the trophic mode is not changed, the VOC profiles can still be adapted to the abiotic and biotic environment[41,56]. Taken the inhomogeneous soil environment and all the factors that can influence fungal activity and fVOC profiles[53–56], it is possible that a laboratory setup in which the effectiveness of pure compounds, or one-to-one interaction, is studied in closed compartments[35] does not reflect a real interaction scenario in nature. Micro-organisms may, moreover, take volatiles up eventually using them as substrates in their own metabolisms[82]. Such VOC-uptake may quench signaling cues and interfere with various interspecific or interkingdom interactions that are essential to initiate and maintain various interorganismic relationships[20–24]. In future, the importance of fVOCs in fungal ecology should be elucidated under diverse abiotic and biotic environments, and in more natural experimental set-ups. More knowledge about the plasticity of guild- or lifestyle-related fVOC patterns in different environments could be a breakthrough for the chemical ecology of fungi and their economic or agricultural applications. With the present work, we have made the first attempt to understand general fungal volatile chemotaxonomy, and also to provide a computational insight into the characterization of fungal functions based on their VOC profiles. Our data are another cornerstone on which we can build to uncover the functions of fungal VOCs in different ecosystems.

## Methods

**Fungal species and cultivation**. The forty-three fungal species studied herein are listed in Supplementary Table S1[51]. For the VOC analysis, the fungi were cultivated in glass cuvettes (7 cm diameter and 6.6 cm depth, total volume approx. 254 mL) on modified synthetic Melin–Norkrans medium (containing (L$^{-1}$) 10 g glucose, 2.5 g NH$_4$-Tartrat, 0.5 g KH$_2$PO$_4$, 0.25 g (NH$_4$)$_2$SO$_4$, 0.15 g MgSO$_4$ × 7H$_2$O, 0.05 g CaCl$_2$, 0.025 g NaCl, 1 mL FeCl$_3$ (1% (w/v)), 100 μL thiamine HCl (0.1% (w/v)), and 1% (w/v) Gelrite, pH 5.2)[18] and cultivated under controlled dark conditions at a constant temperature of 23 °C[41]. For each species, the exponential growth phase was determined (the growth curves are shown in Supplementary Fig. S4) and the emission measurements were always started at the beginning of the exponential hyphal growth phase[18], to ensure comparability. This phase was chosen as most of the secondary metabolites of fungi are shown to be formed during this developmental stage[23,83], i.e., after completion of the initial growth and immediately before the transition to the next developmental stage (which is sporulation for most of the fungi). At the end of each experiment, the Petri dishes were scanned and the area of the fungal mycelia was determined using ImageJ software (Rasband, ImageJ, U. S. National Institutes of Health, Bethesda, Maryland, USA, https://imagej.nih.gov/ij/, 1997–2016).

**Online fVOC measurements by PTR-ToF-MS**. VOC emissions were measured with the previously described platform[18], consisting of 14 cuvettes and a connected proton transfer reaction - time of flight - mass spectrometer (PTR-ToF-MS 8000, Ionicon Analytik GmbH, Innsbruck, Austria). The cuvettes were supplied with VOC-free air under ambient CO$_2$ concentration employing a gas calibration unit (GCU, Ionicon Analytik GmbH, Innsbruck, Austria)[18]. The fVOCs were measured for 48 h each, switching between the cuvettes sequentially. The gas composition of each cuvette was recorded with PTR-ToF-MS for 5 min. Within this time the cuvette air was exchanged completely and a steady state was reached[18]. Between the measurements, the measuring tubes were flushed with VOC-free air by switching for 10 s to the completely empty background cuvette.

PTR-ToF-MS raw data were collected and analyzed following the routine procedures[84,85] in MATLAB (R2011b; MathWorks, Natick, MA, USA). The calculated signals in counts per second (cps) were normalized to 10$^6$ reagent ion counts to account for differences in the absolute humidity in the different cuvettes[18]. The normalized counts per second (ncps) were calculated and the data averaged. Finally, the data were normalized to the respective area of fungal mycelium and the circa 70 min accumulation time. The compounds detected by PTR-ToF-MS (Supplementary Table S2[51]) were, in addition, converted from ncps to pmol (cm$^{-2}$ h$^{-1}$) when the sensitivity (ncps/ppbv) of the specific compound was known or could be estimated. PTR-ToF-MS sensitivities were derived from

calibration curves by measuring a mix of VOC standards passing through the whole system and an empty cuvette (Supplementary Table S2[51]).

The signals detected by pure growth medium (background) were subtracted from the samples with mycelium. For this purpose, cubic splines were laid through the averaged background signals during the entire measurement. The intensities of the interpolated signals were then subtracted from all measurements with fungi. All $m/z$ attributable to isotopologues (containing $^{13}C$, $^{18}O$) were also removed.

The non-targeted PTR-ToF-MS measurements poses a challenge for the accurate identification and quantification of the detected masses. The annotations are based on previously published data from PTR-ToF-MS-based fungal/microbial and plant VOC analyses, soil matrix, and the mVOC database (Supplementary Table S2[51,86]). Following, we determined the molecular formulae based on accurate mass measurements and detections of the corresponding naturally occurring isotopes. Mass features that are likely originating as fragments of a (related) compound are given in Supplementary Table S2[51] and were determined by correlation analysis ($R^2 > 0.9$) using "ToF data plotter". Using a combination of the above strategies, we were able to assign the mass features to 56 different molecular formulae (Supplementary Table S2[51]).

**Offline fVOC measurements by GC–MS**. After completion of PTR-ToF-MS analysis, VOCs were collected for further 16 h on polydimethylsiloxane (PDMS) coated stir bar "twisters" (Gerstel GmbH, Mülheim an der Ruhr, Germany). Subsequently, the VOCs were analyzed by thermal desorption (TDU, Gerstel)—gas chromatography–mass spectrometry (GC type: 7890A; MS type: 5975C, Agilent Technologies, Palo Alto, CA, USA) using a capillary GC column ((14%-Cyano-propyl-phenyl)-methylpolysiloxane; 70 m × 250 μm, film thickness 0.25 μm; Agilent J&W 122-5562 G, DB-5 MS +10m DG). The samples were desorbed with a temperature gradient from 40 to 300 °C, followed by a holding time of 6 min. The compounds were refocused on Tenax at −60 °C and desorbed to 325 °C at a rate of 12 °C s$^{-1}$, after which a holding time of 2 min was applied. As a carrier gas-liquid nitrogen with a constant flow rate of 1 mL min$^{-1}$ was used. The initial temperature of GC oven was 40 °C. At first, the temperature was increased at a rate of 10 °C min$^{-1}$ to 130 °C at a rate of 10 °C min$^{-1}$, holding for 5 min. After this the temperature was increased in following steps: to 175 °C at a rate of 80 °C min$^{-1}$, holding for 0 min; to 200 °C at a rate of 2 °C min$^{-1}$, holding for 0 min; to 220 °C at a rate of 4 °C min$^{-1}$, holding for 0 min and finally to 300 °C at a rate of 100 °C min$^{-1}$, holding for 6 min.

The chromatograms were analyzed using the Enhanced ChemStation software (MSD ChemStation E.02.01.1177, 1989–2010 Agilent Technologies, Santa Clara, CA, USA). Compound identification was based on the comparison of the representative masses using the National Institute of Standards and Technology (NIST) Mass Specral Library (NIST 11) and Wiley 275 GC/MS Library (Wiley, New York), and finally confirmed by comparison of the Kovats retention indices. Kovats retention indices were calculated based on chromatography retention times of a saturated alkane mixture (C9 – C25; Sigma-Aldrich, Taufkirchen, Germany). The potential changes in the GC–MS sensitivity were corrected by normalizing to the internal standard (monoterpene δ-2-carene). The compounds were quantified using the external standards: sabinene, α-pinene, linalool, methylsalicylate, β-caryophyllene, α-humulene, geraniol, and bornylacetate[18,41].

To consider differences in mycelium biomass among the different fungal species, emission rates were normalized to the area of fungal mycelium.

### Statistical analyses

*Discriminant analysis of principal components (DAPC)*. DAPC was applied to infer the phylogenetical and fungal guild-based variations using the complete emission profiles[87,88]. Two different, alternative groupings to fungal guilds were used for the DAPC analysis (trophic mode and optional trophic mode as shown in Supplementary Table S1[51]). Data were centered and scaled to unit variance before performing DAPC, to ensure equal weighing between PTR-ToF-MS and GC–MS data set[89,90]. DAPC was implemented in the *adegenet* package v2.1.1[91] in R[92]. To avoid unstable assignments of individuals to clusters, the number of retained principal components (PCs) was determined by cross-validation using "xvalDapc" function[93] in *adegenet* package. The number of retained PCs in the DAPC model associated with the lowest Mean Squared Error.

*Calculation of the phenotypic integration of fVOCs across trophic modes and structural chemical classes*. To calculate the PI we used a method previously applied by Junker et al.[50] to assess the level of covariation of compounds emitted by all samples, by samples originating from each of the trophic mode, as well as of compounds of structurally related compound classes within the trophic modes and across all samples. In brief: For each of these data sets, we determined the Pearson's correlation coefficient $r$ for all pairs of compounds and calculated eigenvalues of the resulting correlation matrix. The variance of the eigenvalues gave the integration index, a measure of the magnitude of PI. To correct for varying numbers of samples within each data set, the integration index was standardized by subtracting the expected value of integration under the assumption of random covariation (random covariation = (number of compounds emitted by the species – 1)/number of samples[94,95]) and then dividing by the potential maximum value of PI in the given data set. Finally, the results were multiplied by 100 to obtain the percentages. To test whether PI-values deviate from a random expectation, we calculated PI of

$n = 10,000$ randomly drawn from the data set. PI-values larger than the 95% quantile of the $n = 10,000$ random PI-values were considered significantly higher than expected by chance.

*Machine learning strategy for biomarker discovery*. To identify the key volatile compounds, (i.e., the biomarkers) that can distinguish phylogenetic and functional guild clusters, we have developed a strategy for ensemble machine learning (ML) algorithms that avoids the potential bias of a given data set by different ML models[96]. This strategy takes advantage of both Wrappers and Embedded Methods[97], implemented by the random forest- (rf)[98] and Bagged CART-based recursive feature elimination algorithm (wrapper method, using "rfe" function in the caret package in R) and regular random forest (embedded method, using "train" function in the caret package[99]) in R[92].

To avoid the possible interactive influence of scale and sparsity differences between PTR-ToF-MS and GC–MS data sets, we trained the ML models separately for each data set[18,100]. The data were randomly divided into a training set (75%) and a test set (25%). The training set was used to train the models, and the performance of the models was independently tested with those samples composing the test set. To mitigate the potential problem resulting from class imbalance, we randomly sampled (with replacements) the minority class to be the same sample size as the majority class of the training set (using "upSample" function in caret package), while the test set remained original. Ten-fold cross-validation was performed to optimize the classification model at each iteration. Hyperparameters were tuned using "tuneLength" function in the caret package to reach a stable and optimal model accuracy. All those configurations together enabled us to achieve a very high model performance while avoiding overfitting.

In wrapper learning, the "rfe" function returns an integer corresponding to the optimal subset size. After determining the optimal subset size, the "selectVar" function was used to calculate the best rankings for each variable across all the resampling iterations. Each model ultimately yielded a list of compounds corresponding to the optimal subset size (predictors subset).

Finally, we trained the final models using the predictor subsets (top 15 predictors) to derive the significance of the final predictor and yield accuracy, sensitivity, and *p*-value of the model accuracy for the final model. To minimize potential model bias and false-positive biomarker candidates, the predictors consistently selected by all three models were considered the final biomarkers. The same predictors could have a different weight in the different models. When this was the case, the mean values of their ranks in the different models were calculated, resulting in the final ranks of the biomarkers. In the selection of biomarkers, insignificant models and models with accuracy lower than 80% were excluded. For color visualization, the data were z-score standardized to have zero means and standard deviation of 1.

**Statistics and reproducibility**. Sample size was based on prior experience assuming comparable variation in the resulting data[18]. The VOC measurements are based on distinct, biologically independent samples and no data was excluded. The number of replicates is given in respective figure legends. Different fungal species and individual samples were randomly distributed in the available cuvettes.

**Reporting summary**. Further information on research design is available in the Nature Research Reporting Summary linked to this article.

## Data availability
The data sets generated during and/or analyzed during the current study are either available in supplementary materials (larger data sets deposited in public repository[51]) or from the corresponding author upon reasonable request.

## Code availability
All within this study applied R codes are available at the public repository Zenodo[101].

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

## Acknowledgements

We thank Felix Antritter and Baris Weber (Helmholtz-Zentrum München, Germany) for technical assistance with the cuvette platform and the GC–MS analysis, and Corina Vlot (Helmholtz Zentrum München, Germany) and Monika Schmoll (Austrian Institute of Technology, Austria) for providing several fungi (for species list please see the Supplementary Table S1[51]) from their collections. We thank the China Scholarship Council (YG) (grant number: 201608130095) and the Deutsche Forschungsgemeinschaft DFG (MR, AP and JPS) (grants: RO5311/4-1, PO361/20-2 and SCHN653/5-1, 5-2) for funding.

## Author contributions

M.R., A.P., K.P. and J.-P.S. designed the study with the contribution of F.W. and P.B. Y.G. performed the experiments supported by F.W. and J.P.B. Y.G. analyzed the PTR–MS data together with W.J. Y.G. analyzed the GC–MS data together with A.G. and run all the statistical analyses. R.J. calculated the phenotypic integration of fVOCs. Y.G. prepared the figures and tables and wrote the manuscript together with M.R. and J.-P.S. All authors contributed to data analysis, interpretation of the findings, and edited and approved the manuscript.

## Competing interests

The authors declare no competing interests.
