## [Transparent Peer Review File · Communications Biology]

Reviewers' comments:

Reviewer #1 (Remarks to the Author):

Guo et al. present very interesting new results on phenotyping fungal diversity. They try to understand the links between the fungal VOC signatures to trophic modes and ecological functions. The paper is well written and would be a valuable and novel addition to the existing mVOC literature in the context of fungal chemical ecology and beyond. I would have relatively minor suggestions and would recommend the paper is published if they can be addressed in the revision:

1) The authors give a great example of how fungal VOCs can be used to explain fungal diversity and gain taxonomical information (beautifully shown by the figures and the analysis) using different analytical and statistical approaches. This gives a good but perhaps a somewhat "static" picture. VOC profiles should not be static but are expected to change over the time of growth so what I am wondering about is the dynamic part of the big picture. Specifically I wonder how the phenotypes and trophic functions change over time. It is not a criticism but the online cuvette measurements by PTR-MS should have been perfectly suited to making an insight also into the time-resolved factor profiles changing as a function of fungal growth so I am curious if this is something you have looked at.

2) I really like Figure 1. The clustergram shows very unique patterns. However, it is interesting or perhaps a little surprising that all these patterns are so unique. I was expecting a little greater similarities in the common pathways shared by all the taxa and chemical components of primary and secondary metabolisms. It would be much more informative if each taxon on the y axis and each compound on the x axis were shown. Otherwise, the similarities on the dendrograms cannot be easily followed.

3) One general challenge in the microbial VOC community is how to perfectly standardize fungal and bacterial phenotypes and be systematic in recording and comparing chemical fingerprint databases. It is great to see the authors highlight the need for this and propose consistent standardization. Area of mycelia and time of growth make sense but is there a reason why 23 C temperature was chosen? Is it suggested that this temperature is used for standardization instead of 25 C? Another general question is whether using constant temperature could favor certain taxa more adapted to this temperature compared to those that grow better at lower or higher temperature.

4) The combined PTRTOF and GC data look beautiful overall but an arbitrary emission unit $\text{ncps cm}^{-2} \text{s}^{-1}$ unit can be confusing for many readers unfamiliar with PTR-MS. Sensitivities are provided which in principle could allow a reader to convert ncps to ppb and then to emission rates. I think it unnecessarily increases mental processing time for readers where it should be trivial for the authors to convert all emission units consistently to $\text{pmol cm}^{-2} \text{h}^{-1}$. The sensitivities and uncertainties for uncalibrated compounds could be estimated from transmission and a combination of known and default reaction rate constants (e.g. Holzinger et al., 2019). I strongly suggest the authors make this effort if that is possible because the absolute emission rates will also be useful in referencing in future follow-up studies.

5) The acronym fVOC which was also used in your previous paper makes sense to emphasize fungal origin of VOCs. However, it would be nice at least to say in introductory sentence that fVOCs are part of the broader class of microbial VOCs (mVOCs) especially that there is a large relevant mVOC literature base that is not cited (e.g. Quin et al., 2014; Uehling et al., 2017).

6) I am praising the excellent work of the authors to elucidate individual compounds and segregate them into classes. Benzenoids and furanoids seem particularly fascinating to me because they are expected from converting polysaccharides in plant lignin and hemicellulose. The class I was expecting to see but which was missing are sulfur containing compounds (e.g. DMS, DMDS). These VOC ions were missing in Table S2 (e.g. only the glycol on m/z 63 but no m/z consistent with DMS). This makes

me wonder if the detection of sulfur compounds was compromised in the analytical system, if these compounds may have been below the detection limit or if the sulfur compounds were for some reason not emitted by these fungi. Another important classes are fatty acids and fatty acid esters. These compounds were previously reported from symbiotic mycorrhizal fungal endosymbionts are quite common, for instance, they commonly affect certain mycorrhizal fungi (Uehling et al., 2017). It would be great to follow up in this paper on the potential role of this class in the symbiosis.

7) It is surprising to see diethyl phthalate (DEP) emitted by fungi. This compound is a known anthropogenic pollutant and a plasticizer so the question is if it may have come from the use of materials in the cuvette or the sampling system? The reference in Table S6 points to a composting paper but there DEP was used as an anthropogenic substrate not as fungal emission. Do the authors know of any source indicating fungal production of DEP and what would be the metabolic precursor for this compound?

8) Figure 2, nicely shows the complementary power of PTR-TOF-MS and GC-MS. The section in line 144 could be made a little more balanced in elucidating complementary values of the GC and PTR-MS techniques. The text "In general, the fVOC patterns detected by GC-MS showed a higher chemical diversity than the patterns detected by PTR-MS as visualized in Figure 2 by the number of detected compounds." suggests that GC showed a higher chemical diversity than PTR-MS but Figure 2 generally shows a factor of 2-3 higher number of VOC ions in PTR-MS than GC. Did you mean higher variability? I would generally suggest to balance this section more, especially that other recent PTRTOF 8000 measurements generally report in the order of hundreds of VOC ions, and the most recent PTR-MS (PTR3, Vocus) in the order of thousands and therefore should even better complement the speciated but less time-resolved GC measurements.

9) I was missing the discussion of parameters that were not measured but that could affect the results or be confounding variables. For example was pH measured or controlled?

10) The number of replicates seems good, but I wonder how consistent the results were because it would be interesting to learn more about the natural variability of emissions over time as some compounds might not replicate well if they are potentially used in short-term signaling.

11) I am just curious if CO₂ respiration was measured and what was the concentration of CO₂ in the zero air source supplied to the cuvettes.

12) I was missing in the concluding remarks setting the stage for future progress in these important directions. For example, how important is the role of gaseous substrates and how the uptake chemotypes would look like for the same fungal species?

13) L166, is the use of "odor profiles" appropriate? Some VOCs may not have olfactory functions so the use of "VOC profiles" may be more appropriate.

14) L413 Add the amount of trimming between valve switching. Did the signal of SVOCs reached steady state (due to switching not, in the cuvette).

Technical

15) L315. Replace the first comma ",", with the full stop "."

References:

Holzinger, R., Acton, W. J. F., Bloss, W. J., Breitenlechner, M., Crilley, L. R., Dusanter, S., Gonin, M., Gros, V., Keutsch, F. N., Kiendler-Scharr, A., Kramer, L. J., Krechmer, J. E., Languille, B., Locoge, N.,

Lopez-Hilfiker, F., Materić, D., Moreno, S., Nemitz, E., Quéléver, L. L. J., Sarda Esteve, R., Sauvage, S., Schallhart, S., Sommariva, R., Tillmann, R., Wedel, S., Worton, D. R., Xu, K., and Zaytsev, A.: Validity and limitations of simple reaction kinetics to calculate concentrations of organic compounds from ion counts in PTR-MS, *Atmos. Meas. Tech.*, 12, 6193–6208, <https://doi.org/10.5194/amt-12-6193-2019>, 2019.

Quin, M.B., Flynn, C.M. and Schmidt-Dannert, C., 2014. Traversing the fungal terpenome. *Natural product reports*, 31(10), pp.1449-1473.

Uehling, J., Gryganskyi, A., Hameed, K., Tschaplinski, T., Misztal, P.K., Wu, S., Desirò, A., Vande Pol, N., Du, Z., Zienkiewicz, A. and Zienkiewicz, K., 2017. Comparative genomics of *Mortierella elongata* and its bacterial endosymbiont *Mycoavidus cysteinexigens*. *Environmental microbiology*, 19(8), pp.2964-2983.

Reviewer #2 (Remarks to the Author):

As someone who works with fungal traits and VOCs, I was very eager to read this paper. Authors characterized the fungal volatilome of 43 different fungal species from different functional groups. Authors then proceeded to compare and contrast volatile profiles among species and functional groups. They found that the volatilome varies by species but found similarities at the functional group level and broad taxonomical level (i.e. phylum). Moreover, authors tested their assumptions using machine learning and found that functional group could be predicted based on the volatilome. Overall, this is an outstanding paper linking volatile profiles with ecological functions in fungi.

I have a few comments:

1. Authors determined functional grouping based on literature which is not standardized and can provide biased results. I recommend authors to compare their functional grouping with FUNguild. They can use genbank to get accession numbers for the ITS sequence of the species they worked with and run them on FUNguild.
2. The functional grouping mycoparasitic and pathogenic are confusing. Mycoparasites can be pathogens of fungi. And many pathogens can be parasites before infecting. I would like to see a broader explanation on what the authors' rationale was to use these functional grouping. Perhaps pathogenic fungi refer only to plant and animal pathogens? Fig 6 suggest plant pathogens, if so, please change to plant pathogens on figures and in the manuscript. Alternatively, authors could strengthen their results by showing FUNguild trait-based grouping as suggested above.
3. Authors say they grew fungi in standard conditions, but what does that mean? I would assume standard conditions for saprotrophic and mycorrhizal are very different. I have never worked with mycorrhizal fungi so I am not entirely sure how they are cultured but it is my understanding that they cannot grow without their host. Can you please elaborate if this was the case for the mycorrhizal species you had, and if so, how did you make sure that the trapped VOCs for mycorrhizal were exclusively from the fungi and not from their host.

Line-specific comments:

398-399 - Can you please add briefly what the composition of the medium is?

441-442 - What does the projected area of the fungal mycelium mean? And how did authors do this?

279-281 - very interesting!

861 - please elaborate on the methods how z-scores were calculated.

Reviewer #3 (Remarks to the Author):

2020 Guo et al Volatile chemical diversity across fungal taxa and lifestyles

The authors present a thorough investigation of fungal volatilomes, to my knowledge the most comprehensive study for fungi. The authors attempt to understand fungal chemotaxonomy and were able to show, based on their fundamental analyses, that distinct VOC profiles relate to trophic modes, life style, substrate-use and host-type of the fungi.

Due to the lack of available methodology and analytic capacity in the past such a large systematic approach was not possible. The authors had the appropriate equipment and a group of experts to initiate this extensive study. The authors present a very well-done analysis of volatilomes of 43 fungal species. These 43 fungi were selected because they represent different lifestyles, various phyla, classes and families, inhabit different hosts, possess different metabolic pathways and capabilities. This study has the advantage to use two analytical methods to determine the volatilomes of the fungi, many earlier studies used only one method, subsequently the VOC spectra were less complex. Therefore, the authors of this study base their studies on a very broad and fundamental data set.

Finally, an approach using machine learning was set up to identify a set of characteristic VOCs (finger prints), 15 predictors with 80% accuracy were found, nicely summarized in Fig 6.

The whole paper is well written and therefore easy to follow for the reader.

I have no suggestions/comments to improve the manuscript and/or the figure presentations.

The manuscript can be published as it is!

Dear Editor, dear Reviewers,

Thank you for all the corrections, comments and suggestions on our manuscript. Please find below our point-by-point response to the issues raised by the reviewers. The changes have been made accordingly in the manuscript, and can be followed by yellow highlighting.

Reviewers' comments:

Reviewer #1 (Remarks to the Author):

Guo et al. present very interesting new results on phenotyping fungal diversity. They try to understand the links between the fungal VOC signatures to trophic modes and ecological functions. The paper is well written and would be a valuable and novel addition to the existing mVOC literature in the context of fungal chemical ecology and beyond. I would have relatively minor suggestions and would recommend the paper is published if they can be addressed in the revision:

1) The authors give a great example of how fungal VOCs can be used to explain fungal diversity and gain taxonomical information (beautifully shown by the figures and the analysis) using different analytical and statistical approaches. This gives a good but perhaps a somewhat "static" picture. VOC profiles should not be static but are expected to change over the time of growth so what I am wondering about is the dynamic part of the big picture. Specifically, I wonder how the phenotypes and trophic functions change over time. It is not a criticism but the online cuvette measurements by PTR-MS should have been perfectly suited to making an insight also into the time-resolved factor profiles changing as a function of fungal growth so I am curious if this is something you have looked at.

Answer: To give a more detailed insight in variability of the emission profiles, we now report the 48h means with standard deviation for each sample in the new Supplementary Table S4c. From these data it is visible that during the short measuring period of 48 hours the VOC emission profiles remain rather stable for most of the species / compounds. Although we agree with the reviewer's suggestion that studying dynamics of fungal emissions is very interesting, we have difficulties to see how to further integrate such data for each compound (61 compounds) and for each species (43 species) into the storyline of the present manuscript (without drastically extending the length of the manuscript). Previously, we have already presented more detailed time series data for selected fungi (i.e. *Trichoderma* species) in the publication Guo et al. (2020*) together with the setup of the cuvette system applied also herein. The focus of the present work is, however, to highlight the differences and similarities in VOC profiles between taxa and trophic modes based on two types of MS data, GC-MS and PTR-MS. The diversity of fungal VOCs we present herein is based on the fact that the collected data (PTR-MS and GC-MS) complement each other. The detailed presentation of the temporal VOC data from the PTR-MS measurements would only cover about half of the emission data and rather the chemically simpler compounds. In our opinion, dynamic emission curves would not add an additional value for the conclusion and discussion at this stage, but would rather move the focus of the central statements of the present work. Moreover, in our opinion, detailed insights into such emission dynamics are of interest

especially when the fungus is developing rapidly (e.g. *Trichoderma* in Guo et al., 2020*) or is exposed to environmental changes during the measurement period. This was not the case here: Most of the fungal species studied herein, were slow-growing under the chosen conditions and/or did not show dramatic changes in their developmental stage on the plates within the given measurement time. Taking these facts, and in order to maintain the focus of the present work, we would like to restrain from adding several additional figures to visualize emission dynamics over the measurement period.

*Guo, Y., Jud, W., Ghirardo, A., Anritter, F., Benz, J.P., Schnitzler, J.P. *et al.* Sniffing fungi—phenotyping of volatile chemical diversity in *Trichoderma* species. *New Phytol.* 227, 244-259 (2020).

2) I really like Figure 1. The cluster gram shows very unique patterns. However, it is interesting or perhaps a little surprising that all these patterns are so unique. I was expecting a little greater similarities in the common pathways shared by all the taxa and chemical components of primary and secondary metabolisms. It would be much more informative if each taxon on the y axis and each compound on the x axis were shown. Otherwise, the similarities on the dendrograms cannot be easily followed.

Answer: Following this valuable suggestion, we have now created a new heat map in which we have sorted the fungi according to their taxa and the compounds according to their chemical structures. We find that this new figure complements the results shown in Figure 2 very well, which is why we have decided to link the new figure to Figure 2 as supplemental material (new Supplemental Fig. S1). For the sake of clarity, we do not show in the new figure individual compounds, but chemical groups, similar as done in Figures 2 and 5 of the present paper.

New Supplemental Figure 1

Fig. S1. Heatmap of fungal volatile organic compounds (fVOC) emission profiles from the examined 43 fungal species. The emission intensity of fVOCs grouped in structural classes is shown for each fungal species. The emission rates ($\text{ncps cm}^{-2} \text{s}^{-1}$ and $\text{pmol cm}^{-2} \text{h}^{-1}$) based on PTR-MS and GC-MS data, respectively, are color coded: red indicates high and blue indicates low emission.

3) One general challenge in the microbial VOC community is how to perfectly standardize fungal and bacterial phenotypes and be systematic in recording and comparing chemical fingerprint databases. It is great to see the authors highlight the need for this and propose consistent standardization. Area of mycelia and time of growth make sense but is there a reason why 23 C temperature was chosen? Is it suggested that this temperature is used for standardization instead of 25 C? Another general question is whether using constant temperature could favor certain taxa more adapted to this temperature compared to those that grow better at lower or higher temperature.

Answer: Temperature may indeed change the emission profile of fungi, similar to almost any other change in the environment. We chose 23°C as standard temperature as large part of the herein investigated fungi were previously reported to grow at this, or near to this, temperature under laboratory conditions. It might indeed be that for studying fungal species specialized to specific temperatures, the standard conditions need to be adjusted. This is however the case also for many other specialized microbes that need special growth environment (if they at all can be grown in the laboratory conditions).

We now additionally discuss that temperature may influence the detected emission profile:

L260-264 *“Difficulties arise from restrictions in analytical techniques, limited number of studied species and varying experimental conditions such as physiological stage of the organisms, sampling period, nutrient availability and other environmental factors such as pH or temperature, that all can influence the fungal performance and the fVOC patterns”*

L390-394: *“Taken the inhomogeneous soil environment and all the factors that can influence fungal activity and fVOC profiles^{54,55,56,57}, it is possible that a laboratory setup in which the effectiveness of pure compounds, or one-to-one interaction, is studied in closed compartments³⁵ does not reflect a real interaction scenario in nature.”*

4) The combined PTRTOF and GC data look beautiful overall but an arbitrary emission unit $\text{ncps cm}^{-2} \text{s}^{-1}$ unit can be confusing for many readers unfamiliar with PTR-MS. Sensitivities are provided which in principle could allow a reader to convert ncps to ppb and then to emission rates. I think it unnecessarily increases mental processing time for readers where it should be trivial for the authors to convert all emission units consistently to $\text{pmol cm}^{-2} \text{h}^{-1}$. The sensitivities and uncertainties for uncalibrated compounds could be estimated from transmission and a combination of known and default reaction rate constants (e.g. Holzinger et al., 2019). I strongly suggest the authors make this effort if that is possible because the absolute emission rates will also be useful in referencing in future follow-up studies.

Answer: We now have converted the $\text{ncps cm}^{-2} \text{s}^{-1}$ to $\text{pmol cm}^{-2} \text{h}^{-1}$ as suggested by the reviewer.

Where feasible, we used instrument sensitivities to pure VOC standards or calculated the sensitivities of compounds by taking into account the molecular masses, functional groups, polarity and reaction rate constants. For some ion masses we still refrained from converting the signals to actual emission rates, namely, if the ion is attributable to a fragment of several different compounds, e.g. m/z 41.038. Many of the measured masses are fragments of compounds that remained unknown and could only roughly be connected to a chemical class based on the exact masses (Supplementary Table S2). Thus, the conversion to pmol is not trivial and for certain compounds the instrument sensitivity was roughly estimated to the best of our knowledge, potentially leading to substantial uncertainties of the quantification. For this reason, we wish to show both data, ncps and pmol stating in the legend of the latter how the data was converted. All the emission rates (raw data and means + sd in pmol and ncps) are given now in the modified Supplementary Table S4a-c.

5) The acronym fVOC which was also used in your previous paper makes sense to emphasize fungal origin of VOCs. However, it would be nice at least to say in introductory sentence that fVOCs are part of the broader class of microbial VOCs (mVOCs) especially that there is a large relevant mVOC literature base that is not cited (e.g. Quin et al., 2014; Uehling et al., 2017).

Answer: We now state in the introduction that fVOCs are part of the broader class mVOCs, as suggested by the reviewer. We now write:

L65-66: *“Fungi and other micro-organisms emit a wealth of highly diverse volatile organic compounds (microbial VOCs; mVOCs)”*

We also now discuss and cite the proposed interesting literature as follows:

L306-307 “Regarding potential biological functions of guild- and taxa-related fVOCs, sesquiterpenes are a prominent group of molecules with very high structural diversity⁶⁰”

L348-353 “Certain genes that regulate and control the biosynthesis of VOCs may be switched off in monoculture, while the same fungus in mixed culture⁷⁵ or under natural conditions (e.g., in soil) in interaction with other organisms may have completely different emission patterns. Such interactions include also fungal endobacteria, that are ubiquitous⁷⁶ and have been shown to affect the detected fVOC patterns⁷⁷. ”

6) I am praising the excellent work of the authors to elucidate individual compounds and segregate them into classes. Benzenoids and furanoids seem particularly fascinating to me because they are expected from converting polysaccharides in plant lignin and hemicellulose. The class I was expecting to see but which was missing are sulfur containing compounds (e.g. DMS, DMDS). These VOC ions were missing in Table S2 (e.g. only the glycol on m/z 63 but no m/z consistent with DMS). This makes me wonder if the detection of sulfur compounds was compromised in the analytical system, if these compounds may have been below the detection limit or if the sulfur compounds were for some reason not emitted by these fungi. Another important classes are fatty acids and fatty acid esters. These compounds were previously reported from symbiotic mycorrhizal fungal endosymbionts are quite common, for instance, they commonly affect certain mycorrhizal fungi (Uehling et al., 2017). It would be great to follow up in this paper on the potential role of this class in the symbiosis.

Answer: Thank you for considering our work excellent! We have now re-checked the presence of sulphur containing compounds in the studied species and report that these were under detection limit in the present study. Moreover, according to mVOC database (<https://bioinformatics.charite.de/mvoc/>; Lemfack et al., 2017*) microbial DMS and DMDS have previously been mostly shown in the emission profiles of various bacteria or they are emitted by specific fungal species such as *Tuber* spp. that were not studied here. It may, however, be that the herein studied species also release detectable levels of some sulfur containing compounds in different environment.

Regarding fatty acids and fatty acid esters, also these compounds were under detection limit in the present study. This might be because in general heavy, highly oxygenated compounds fall below the detection limit. For such “sticky” compounds one would have to sample for longer times from the same, continuous source, or heat the sample lines with the risk of compound degradation. None of the approaches were taken in the present experiments thus excluding putative emissions of heavier compounds.

*Lemfack, M.C., Gohlke, B.-O., Toguem, S.M.T., Preissner, S., Piechulla, B., & Preissner, R. mVOC 2.0: a database of microbial volatiles. Nucl. Acid Res. 46, D1261-D1265 (2017).

7) It is surprising to see diethyl phthalate (DEP) emitted by fungi. This compound is a known anthropogenic pollutant and a plasticizer so the question is if it may have come from the use of

materials in the cuvette or the sampling system? The reference in Table S6 points to a composting paper but there DEP was used as an anthropogenic substrate not as fungal emission. Do the authors know of any source indicating fungal production of DEP and what would be the metabolic precursor for this compound?

Answer: Putative DEP was indeed detected from two fungi in our study. This compound was not detected in the control treatment (pure medium), so it seems to not to originate from the sampling system. Previously bacteria have been shown to produce this compound and it also was shown that DEP possess antifungal properties (Campos et al., 2010; Liu et al 2018*). Thus it seems likely that microorganisms are able to synthesize it and that it also has potential ecological importance in antagonism. We have now added the reference Campos et al., 2010 to the Supplementary Table S6. In addition, we added the word “putative” to the annotation of this compound as we are at the moment unable to name the possible biosynthetic route in fungi.

*Campos, Vicente Paulo, Pinho, Renata Silva Canuto de, & Freire, Eduardo Souza. (2010). Volatiles produced by interacting microorganisms potentially useful for the control of plant pathogens. *Ciência e Agrotecnologia*, 34(3), 525-535.

Liu et al 2008; Antagonistic activities of volatiles from four strains of *Bacillus* spp. and *Paenibacillus* spp. against soil-borne plant pathogens. *Agricultural Sciences in China*, 9; 1104-1114

8) Figure 2, nicely shows the complementary power of PTR-TOF-MS and GC-MS. The section in line 144 could be made a little more balanced in elucidating complementary values of the GC and PTR-MS techniques. The text “In general, the fVOC patterns detected by GC-MS showed a higher chemical diversity than the patterns detected by PTR-MS as visualized in Figure 2 by the number of detected compounds.” suggests that GC showed a higher chemical diversity than PTR-MS but Figure 2 generally shows a factor of 2-3 higher number of VOC ions in PTR-MS than GC. Did you mean higher variability? I would generally suggest to balance this section more, especially that other recent PTRTOF 8000 measurements generally report in the order of hundreds of VOC ions, and the most recent PTR-MS (PTR3, Vocus) in the order of thousands and therefore should even better complement the speciated but less time-resolved GC measurements.

Answer: We have now rewritten this chapter to be very neutral and balanced. We also now replaced the word “diversity” with “variability” as suggested. We now write:

Lines 145-153: *In general, a higher number of compounds were detected by PTR-MS than by GC-MS from individual species (Figure 2), but the fVOC patterns detected by GC-MS showed a higher variability than the patterns detected by PTR-MS. With PTR-MS especially short-chained alkenes and other carbonyl compounds, that were not possible to measure by the GC-MS set-up, were detected. In contrast, with the Twister/GC-MS-combination mainly sesquiterpenes and other structurally more complex volatile compounds were measured (Fig. 2, Supplemental Tables S2; S3).*

9) I was missing the discussion of parameters that were not measured but that could affect the results or be confounding variables. For example was pH measured or controlled?

Answer: The pH of the medium was 5.2 as mentioned now on line 415. We now discuss also this aspect next to the other variables that may affect the emission from the fungi (please see also our answers to your questions 3, 5 and 12).

10) The number of replicates seems good, but I wonder how consistent the results were because it would be interesting to learn more about the natural variability of emissions over time as some compounds might not replicate well if they are potentially used in short-term signaling.

Answer: We have now calculated the means and standard deviation of the replicates (Supplementary Table 4a) and also means and standard deviation of individual samples over the whole measurement period (Supplementary Table 4c) to give information about the natural variability of the emissions.

11) I am just curious if CO₂ respiration was measured and what was the concentration of CO₂ in the zero air source supplied to the cuvettes.

Answer: The CO₂ concentration in the zero air was following the ambient variations of CO₂ as previously described in Guo et al., 2020. We now mention this on line 431-433. We did not measure CO₂ concentration in the cuvettes. We have, however, planned to improve our measurement system to make also CO₂ measurements in the future.

12) I was missing in the concluding remarks setting the stage for future progress in these important directions. For example, how important is the role of gaseous substrates and how the uptake chemotypes would look like for the same fungal species?

Answer: We discuss now the various aspects that may influence the fVOC release and fVOCs as signaling cues also in the concluding remarks:

L390-399: *“Fungi are highly adaptable and even if the trophic mode is not changed, the VOC profiles can still be adapted to the abiotic and biotic environment^{18,41,57}. Taken the inhomogeneous soil environment and all the factors that can influence fungal activity and fVOC profiles^{54,55,56,57}, it is possible that a laboratory setup in which the effectiveness of pure compounds, or one-to-one interaction, is studied in closed compartments³⁵ does not reflect a real interaction scenario in nature. Micro-organisms may, moreover, take volatiles up eventually using them as substrates in their own metabolisms⁸³. Such VOC-uptake may quench signaling cues and interfere various interspecific or interkingdom interactions that are essential to initiate and maintain various interorganismic relationships²⁰⁻²⁴. In future, the importance of fVOCs in fungal ecology should be elucidated under diverse abiotic and biotic environments, and in more natural experimental set-ups. More knowledge about the plasticity of guild- or lifestyle-related fVOC patterns in different environments could be a breakthrough for the chemical ecology of fungi and their economic or agricultural applications.”.*

13) L166, is the use of “odor profiles” appropriate? Some VOCs may not have olfactory functions so the use of “VOC profiles” may be more appropriate.

Answer: Apologizes for the use of inappropriate wording. We have now exchanged the word “odor” with “VOC” in the whole manuscript.

14) L413 Add the amount of trimming between valve switching. Did the signal of SVOCs reached steady state (due to switching not, in the cuvette).

Answer: We have now added the information about trimming and switching between the cuvettes. We now write:

L435-439 *“The gas composition of each cuvette was recorded with PTR-ToF-MS for five minutes. Within this time the cuvette air was exchanged completely and a steady state was reached¹⁸. Between the measurements, the measuring tubes were flushed with VOC-free air by switching for 10 s to the completely empty background cuvette. For further details, please see the previous work of Guo et al.¹⁸.”*

Technical

15) L315. Replace the first comma “,” with the full stop “.”

Done!

References:

Holzinger, R., Acton, W. J. F., Bloss, W. J., Breitenlechner, M., Crilley, L. R., Dusanter, S., Gonin, M., Gros, V., Keutsch, F. N., Kiendler-Scharr, A., Kramer, L. J., Krechmer, J. E., Languille, B., Locoge, N., Lopez-Hilfiker, F., Materić, D., Moreno, S., Nemitz, E., Quéléver, L. L. J., Sarda Esteve, R., Sauvage, S., Schallhart, S., Sommariva, R., Tillmann, R., Wedel, S., Worton, D. R., Xu, K., and Zaytsev, A.: Validity and limitations of simple reaction kinetics to calculate concentrations of organic compounds from ion counts in PTR-MS, *Atmos. Meas. Tech.*, 12, 6193–6208, <https://doi.org/10.5194/amt-12-6193-2019>, 2019.

Quin, M.B., Flynn, C.M. and Schmidt-Dannert, C., 2014. Traversing the fungal terpenome. *Natural product reports*, 31(10), pp.1449-1473.

Uehling, J., Gryganskyi, A., Hameed, K., Tschapinski, T., Misztal, P.K., Wu, S., Desirò, A., Vande Pol, N., Du, Z., Zienkiewicz, A. and Zienkiewicz, K., 2017. Comparative genomics of *Mortierella elongata* and its bacterial endosymbiont *Mycoavidus cysteinexigens*. *Environmental microbiology*, 19(8), pp.2964-2983.

Reviewer #2 (Remarks to the Author):

As someone who works with fungal traits and VOCs, I was very eager to read this paper. Authors characterized the fungal volatilome of 43 different fungal species from different functional groups. Authors then proceeded to compare and contrast volatile profiles among species and functional groups. They found that the volatilome varies by species but found similarities at the functional group level and broad taxonomical level (i.e. phylum). Moreover, authors tested their assumptions using machine learning and found that functional group could be predicted based on the volatilome. Overall, this is an outstanding paper linking volatile profiles with ecological functions in fungi.

I have a few comments:

1. Authors determined functional grouping based on literature which is not standardized and can provide biased results. I recommend authors to compare their functional grouping with FUNguild. They can use genbank to get accession numbers for the ITS sequence of the species they worked with and run them on FUNGuild.

Answer: Thank you for this suggestion. Actually, the functional grouping was inspired by results from FUNGuild. Most of the trophic modes are kept in our work as suggested by the program. For a few species, however, FUNGuild do not provide very accurate trophic modes. For all *Fusaria* for example the program suggests following: "Pathotroph-Saprotroph-Symbiotroph". Some species were also rather directed to "an alternative" trophic mode than to the "primary" trophic mode. Such a species was e.g. *Heterobasidium annosum*, a well-known plant pathogen, that was directed to the group of saprotrophs by the program. Due to these discrepancies we manually re-checked the groupings based on the actual literature. To be completely transparent, we now added additional row to the Supplementary Table S1 showing the initial grouping by FUNGuild.

We additionally have now used the recently published new database, Fungal Traits (Pölme et al., 2020*), to group our fungal species to trophic modes. This new database, which is a combination of FUNGuild, Fun^{Fun} and additional expert knowledge, grouped the species in the present study to the exactly same trophic modes as done by us manually. Thus, the results from this database verify the correctness of our manually adjusted groupings.

***Pölme, S., Abarenkov, K., Henrik Nilsson, R. et al. FungalTraits: a user-friendly traits database of fungi and fungus-like stramenopiles. *Fungal Diversity* 105, 1–16 (2020).**

2. The functional grouping mycoparasitic and pathogenic are confusing. Mycoparasites can be pathogens of fungi. And many pathogens can be parasites before infecting. I would like to see a broader explanation on what the authors' rationale was to use these functional grouping. Perhaps pathogenic fungi refer only to plant and animal pathogens? Fig 6 suggest plant pathogens, if so, please change to plant pathogens on figures and in the manuscript. Alternatively, authors could strengthen their results by showing FUNguild trait-based grouping as suggested above.

Answer: We agree with the reviewer and have re-worded pathogen to phytopathogen, where applicable. We also show now the FUNGuild based grouping in the Supplementary Table S1 (please see also our answer to your previous comment)

3. Authors say they grew fungi in standard conditions, but what does that mean? I would assume standard conditions for saprotrophic and mycorrhizal are very different. I have never worked with mycorrhizal fungi so I am not entirely sure how they are cultured but it is my understanding that they cannot grow without their host. Can you please elaborate if this was the case for the mycorrhizal species you had, and if so, how did you make sure that the trapped VOCs for mycorrhizal were exclusively from the fungi and not from their host.

Answer: Thank you also for this comment. Several previous studies have shown that fVOC profiles can be dramatically adjusted under specific environmental conditions (Misztal et al., 2018; Guo et al., 2019*). The conditions in the present study were standardized to allow comparison of the species' specific fVOC profiles under specific, controlled conditions. Similar to Guo et al., 2020*, we chose a medium on which all the studied species grow. Regarding mycorrhiza, actually some mycorrhiza can well be cultivated in axenic cultures. Recent studies suggest that mycorrhiza may diverge from saprotrophic clades. At least they still possess several genes related to decomposing abilities enabling them to grow saprotrophically in the absence of a host (Kohler et al. 2015; Miyauchi et al 2020*).

***References:**

Guo, Y., Jud, W., Ghirardo, A., Anritter, F., Benz, J.P., Schnitzler, J.P. et al. *Trichoderma* species differ in their volatile profiles and in antagonism toward ectomycorrhiza *Laccaria bicolor*. *Front Microbiol.* 10, 891 (2019).

Guo, Y., Jud, W., Ghirardo, A., Anritter, F., Benz, J.P., Schnitzler, J.P. et al. Sniffing fungi—phenotyping of volatile chemical diversity in *Trichoderma* species. *New Phytol.* 227, 244-259 (2020).

Kohler, A., Kuo, A., Nagy, L. et al. Convergent losses of decay mechanisms and rapid turnover of symbiosis genes in mycorrhizal mutualists. *Nat Genet* 47, 410–415 (2015).

Miyauchi, S., Kiss, E., Kuo, A. et al. Large-scale genome sequencing of mycorrhizal fungi provides insights into the early evolution of symbiotic traits. *Nat Commun* 11, 5125 (2020).

Misztal, P.K., Lymperopoulou, D.S., Adams, R.I., Scott, R.A., Lindow, S.E., Bruns, T. et al. Emission factors of microbial volatile organic compounds from environmental bacteria and fungi. *Environ. Sci. Tech.* 52, 8272-8282 (2018).

Line-specific comments:

398-399 - Can you please add briefly what the composition of the medium is?

Answer: We have added the information on medium composition on lines 413-415.

441-442 – What does the projected area of the fungal mycelium mean? And how did authors do this?

Answer: We have clarified this part. We now write:

L423- 426: *“In the end of each experiment the petri dishes were scanned and the area of fungal mycelium was determined using ImageJ software (Rasband, ImageJ, U. S. National Institutes of Health, Bethesda, Maryland, USA, <https://imagej.nih.gov/ij/>, 1997-2016).”*

L469-470: *“To consider differences on mycelium biomass among the different fungal species, emission rates were normalized to the area of fungal mycelium.”*

279-281 – very interesting!

861 – please elaborate on the methods how z-scores were calculated.

Answer: We have now added this information to materials & methods.

We now write:

L538-539: *“For color visualization the data were z-score standardized to have zero means and standard deviation of 1”.*

Reviewer #3 (Remarks to the Author):

2020 Guo et al. Volatile chemical diversity across fungal taxa and lifestyles

The authors present a thorough investigation of fungal volatiles, to my knowledge the most comprehensive study for fungi. The authors attempt to understand fungal chemotaxonomy and were able to show, based on their fundamental analyses, that distinct VOC profiles relate to trophic modes, life style, substrate-use and host-type of the fungi.

Due to the lack of available methodology and analytic capacity in the past such a large systematic approach was not possible. The authors had the appropriate equipment and a group of experts to initiate this extensive study. The authors present a very well-done analysis of volatiles of 43 fungal species. These 43 fungi were selected because they represent different lifestyles, various phyla, classes and families, inhabit different hosts, possess different metabolic pathways and capabilities. This study has the advantage to use two analytical methods to determine the volatiles of the fungi, many earlier studies used only one method, subsequently the VOC spectra were less complex. Therefore, the authors of this study base their studies on a very broad and fundamental data set.

Finally, an approach using machine learning was set up to identify a set of characteristic VOCs (finger prints), 15 predictors with 80% accuracy were found, nicely summarized in Fig 6.

The whole paper is well written and therefore easy to follow for the reader.

I have no suggestions/comments to improve the manuscript and/or the figure presentations.

The manuscript can be published as it is!

Answer: Thank you for your kind words! We were very happy to receive this positive feedback.

REVIEWERS' COMMENTS:

Reviewer #1 (Remarks to the Author):

All my comprehensive comments have been satisfactorily addressed by the authors. Thank you very much! The manuscript now looks outstanding! I have no further suggestions and recommend the manuscript is published.

Reviewer #2 (Remarks to the Author):

The authors have addressed all of my concerns and comments. Thank you, very nice paper.